# Trends of Eurasian Perch (*Perca fluviatilis*) mtDNA *ATP6* Region Genetic Diversity within the Hydro-Systems of the Eastern Part of the Baltic Sea in the Anthropocene

**DOI:** 10.3390/ani13193057

**Published:** 2023-09-29

**Authors:** Adomas Ragauskas, Ieva Ignatavičienė, Vytautas Rakauskas, Dace Grauda, Petras Prakas, Dalius Butkauskas

**Affiliations:** 1Nature Research Centre, Akademijos Str. 2, 08412 Vilnius, Lithuania; ieva.ignataviciene@gamtc.lt (I.I.); vytucio@gmail.com (V.R.); petras.prakas@gamtc.lt (P.P.); dalius.butkauskas@gamtc.lt (D.B.); 2Institute of Biology, University of Latvia, Jelgavas Str. 1, LV-1004 Riga, Latvia; dace.grauda@lu.lv

**Keywords:** perch, intraspecific genetic variability, mtDNA *ATP6*, haplogroup, phylogeographic relationships, nuclear power plants

## Abstract

**Simple Summary:**

This study concerns evaluation of the possible negative effects of power plants as one of the most important objects generating environmental pollution in aquatic ecosystems. We attempted to research the intraspecific genetic diversity of a naturally distributed fish species, Eurasian perch, based on a comparison of its mitochondrial DNA sequences. To distinguish naturally occurring mutagenesis from DNA changes caused by thermal or chemical pollution, the molecular data representing patterns of perch populations inhabiting hydro-systems profoundly affected by power plants and non-affected perch populations were collected and analysed. The obtained results indicate that most genetic differences among perch populations representing a large geographic area that encompasses territories from the Baltic Sea to Ukraine could be explained by historical and ongoing natural processes instead of pollution from power plants.

**Abstract:**

The intraspecific genetic diversity of freshwater fish inhabiting hydro-systems of the macrogeographic area spreading from the Black to Baltic Seas requires comprehensive investigation from fundamental and practical perspectives. The current study focused on the involvement of the mtDNA *ATP6* region in the adaptability and microevolution of *Perca fluviatilis* within phylogeographic and anthropogenic contexts. We sequenced a 627 bp fragment encompassing the *ATP6* region and used it for genetic analysis of 193 perch caught in Latvia, Lithuania, Belarus, and Ukraine, representing natural and anthropogenically impacted populations. We evaluated patterns of intraspecific genetic diversity in the *ATP6* region and phylogeographic trends within the studied area compared with previously established D-loop trends. Evaluation of *ATP6* coding sequence variability revealed that among 13 newly detected haplotypes, only two were caused by non-synonymous substitutions of amino acids of the protein. PCoA revealed three genetic groups (I–III) based on the *ATP6* region that encompassed four previously described genetic groups established based on the mtDNA D-loop. The two mtDNA regions (D-loop and *ATP6*) have microevolved at least partially independently. Prolonged anthropogenic impacts may generate new point mutations at the *ATP6* locus, but this phenomenon could be mainly concealed by natural selection and reparation processes.

## 1. Introduction

The 21st century is the peak of the Anthropocene, as all terrestrial and aquatic ecosystems in the world are directly or indirectly affected by various human activities [1,2]. One of the most noticeable anthropogenic activities is associated with the exploitation of energy resources [3,4]. There are different types of power plants used to generate electricity [5], but at the end of 20th century, the most popular were nuclear and thermal power plants (NPPs and TPPs, respectively) [6].

While NPPs are generally considered to be a relatively safe option for generating electricity [3], technogenic accidents like those in Chernobyl [7] and Fukushima [8] show that there is a great trade-off between convenient energy generation and safety of local ecosystems [9] and humankind [10,11]. The accident at the Chernobyl Nuclear Power Plant (ChNPP) in 1986 greatly affected both terrestrial and aquatic ecosystems [12,13]. Even a few decades after the ChNPP accident, the surrounding area, spreading at least 30 km from the epicentre of the ChNPP, remains heavily contaminated due to unacceptable levels of radionucleotides persisting in the environment [13]. The intraspecific genetic variability of some local species inhabiting radiocontaminated areas has increased, while other organisms have not been able to recover or survive, or at least the genetic diversity of affected populations has been significantly reduced [14,15]. In general, the increased radiation, to a smaller degree, affected all of Europe and not only the local Ukrainian and Belarussian ecosystems situated close to the Chernobyl area [12,16]. In addition, there is enough documentation to show that even reasonable exploitation of NPPs causes marked irreversible changes to local ecosystems [17,18], especially when closed-type coolers are used. For example, the effects of such a cooling system on wildlife were evaluated using comprehensive ecological data collected at Lake Drūkšiai during the period of operation of the Ignalina Nuclear Power Plant (INPP) (reviewed by Virbickas and Virbickas [19]). Contrary to NPPs, TPP activities are considered less harmful to the environment (but see [18]). However, many of them still discharge heated water into aquatic ecosystems [17]. Consequently, research regarding the possible anthropogenic impacts on freshwater ecosystems from newly exploited or long-working NPPs and TPPs that utilise water for cooling purposes taken from nearby lakes or rivers is important from practical and fundamental perspectives. Investigations that aim to reveal the trends of changes in aquatic ecosystems, communities, and populations, including evaluating the possible impact on living organisms by studying their genetic diversity level, are also relevant after decommissioning NPPs with closed-type coolers, as these changes, processes, and trends in recovering ecosystems have been poorly studied [19]. To carry out such research, it is necessary to obtain as much information as possible about natural freshwater ecosystems and their components, as it is crucial to distinguish anthropogenic effects from natural historic and microevolutionary processes.

Global freshwater ecosystems and their biodiversity are susceptible to anthropogenic impacts that are not limited to NPP and TPP activities, which affect aquatic abiotic, biotic, and genetic resources [20,21,22]. These ecosystems have great survival, ecological, commercial, and scientific value to humankind [23,24,25]. In these ecosystems, freshwater fish play an important role in food chains [26]. In addition, many freshwater fish species are either commercially exploited [23,27] or keystone species [28]. Even though a large body of data concerning the distribution, abundance, and ecology of many fish species, both in freshwater and brackish environments [29,30], has been collected, the findings on their intraspecific genetic variability [22,31,32,33], phylogeographic relationships [34,35,36], and postglacial history [24,37,38] remain fragmented. Furthermore, such data are quite limited even for extensively studied fish species [20,39]. 

Freshwater ecosystems spreading between the Black and Baltic Sea Regions require special attention, as the ChNPP is located close to the border between Ukraine and Belarus, and this power plant was set approximately at the midpoint between these seas [12,40]. Populations of contemporary aquatic fauna and flora in the Black and Baltic Sea Regions are thought to harbour unique natural diversity because of the Pleistocene [34]. However, during the last ice age, a major qualitative difference between these large territories appeared, as most of the Black Sea Region remained ice free while the Baltic Sea Region—especially its eastern part, including the current territories of Latvia, partially Lithuania, and Belarus—was covered in ice [39]. It is reasonable to use postglacial fish as model species [24] to investigate the trends of microevolutionary processes that occurred during and after the last ice age in the macrogeographic area encompassing these two unique regions. Thus far, there remains a lack of comprehensive genetic diversity studies of postglacial freshwater fish species in the eastern part of the Baltic Sea Region [41,42]. The most recent data show that this region might be distinguished by more complex phylogeographic relationships among aquatic animals than previously anticipated [35,38]. 

Eurasian perch (*Perca fluviatilis* L.) currently occupies a wide natural range that encompasses European (except the Pyrenees, the Apennine Peninsula, northern England, and freshwater systems along the coast of Norway) and Asian (except the Caucasus and Central Asia, southern Mongolia, the Far East from the Amur River, and Siberia) areas [43]. In addition, this predatory species [44,45] can be detected in a few continents because of translocations [35]. Perch and its genetic diversity have been quite intensively studied both within phylogeographic and anthropogenic contexts using various molecular markers. Moreover, studies of its whole genome are steadily increasing [46]. In general, it is important to apply different molecular marker systems to evaluate the genetic variability of perch and reveal different historical, phylogenetic, and microevolutionary processes [47]. Thus far, perch populations have been studied using various mitochondrial DNA (mtDNA) [38,48], DNA microsatellite [49,50,51,52,53], restriction fragment length polymorphism (RFLP) [54,55], random amplified polymorphic DNA (RAPD) [56], amplified fragment length polymorphism (AFLP) [57], major histocompatibility complex (MHC) [58,59], inter-primer binding site (iPBS) [47], and inter-simple sequence repeat (ISSR) [60] molecular DNA markers. Most studies have sequenced the mtDNA D-loop region [35,41,61,62,63] or used DNA microsatellites that are not specific for perch [25,64,65,66]. Just recently, other mtDNA regions, such as *cytb*, *16S* RNA, and cytochrome oxidase subunit I (*COI*), have been included in comprehensive studies [38,67]. Moreover, the number of whole mtDNA genomes that have been sequenced is increasing [46]. The maternally transmitted mtDNA genome [68] of perch is 16,537 bp in length and consists of 37 genes and two non-coding regions [69], like the mtDNA genomes of most other known fish species [70]. In addition, species-specific DNA microsatellites have been created to examine various European and Asian perch populations [71,72,73,74]. The creation and testing of new molecular markers suitable for different applications, especially for studies of intraspecific genetic variability of perch, are still necessary [74,75]. Intraspecific genetic diversity of perch should be investigated more comprehensively within phylogeographic, anthropogenic, conservation, and other important contexts—including the Baltic Sea Region and other regions of Eurasia—to explore theoretical and practical scientific questions regarding the genetics of this species. 

In some fish species, the ATP synthase F0 subunit 6 (*ATP6*) and ATP synthase F0 subunit 8 (*ATP8*) loci are the most rapidly evolving and adaptable loci among mtDNA genes [76,77,78]. In comparison to marine fish species, there has been no serious attempt to evaluate the potential of the *ATP6* marker for freshwater fish species in Europe based on data from intraspecific genetic research. Based on data reported by Vasemägi et al. [46], within the mtDNA genomic context, *ATP6* gene variability is quite similar to that of other coding mtDNA regions. However, to date, *ATP6* has not been used specifically for comprehensive research on intraspecific variability of perch populations in Europe and Asia.

The present study presents trends of the genetic structure and phylogeographic relationships among the studied Eurasian perch populations inhabiting hydro-systems of Lithuania, Latvia, Belarus, and Ukraine. We generated these results using the mtDNA *ATP6* region. We placed special emphasis on geographic distance among perch populations as well as their genetic diversity patterns potentially affected by other geographic factors, such as height above sea level and different types of water bodies. Furthermore, we evaluated the possible effects of anthropogenic activity on natural selection that could be related to the generation of new point mutations at the *ATP6* locus. We also compared the trends of genetic diversity and phylogeographic relationships among perch populations in the studied macrogeographic area revealed by *ATP6* region research with those revealed by mtDNA D-loop fragment studies. Finally, we discuss the potential use of the *ATP6* region to assess anthropogenic impact, especially caused by NPP activities, on the genetic structure of perch. The following hypotheses were tested: the possible anthropogenic effect is reflected as either increased mutability of perch inhabiting water bodies used as coolers of NPPs and TPPs (H1) or as reduced genetic diversity of the studied fish populations in particular locations (H2).

## 2. Materials and Methods

### 2.1. Sampling and Data Treatment

Samples of perch were delivered to the Laboratory of Molecular Ecology, Nature Research Centre (Vilnius, Lithuania) by hydrobiologists and fishermen. A total of 193 perch (both juvenile and adult individuals) were investigated during the current study; the sex and precise age of the individuals were not determined. A detailed list of samples collected from Lithuania, Latvia, Belarus, and Ukraine is shown in Table 1. Samples that were collected in the same geographic locations at different times were combined to form larger spatial samples representing the same hydro-systems. Similarly, the Neris River samples marked as 9a and 9b, as well as other spatial samples represented by less than five individuals, were grouped into larger spatial samples. Therefore, the specimen from the Chernobyl area (11a) was grouped with another sample from Ukraine and treated as one group in all major genetic analyses. Moreover, separate analyses were not conducted on two small Belarusian samples (10c and 10d). The spatial distribution of the samples within the studied macrogeographic area (from the Baltic Sea to Ukraine) represents the phylogeographic context of this investigation. Additional information regarding height above sea level (altitude in metres) of the studied water bodies was also collected (Table 1) and analysed. To evaluate possible trends of the expected genetic differences of samples collected in different types of water bodies, mainly lakes, the studied water bodies were categorised based on available information about their depth and trophic status. 

This study also focused on the anthropogenic context, especially effects associated with NPP and TPP activities. It should be noted that all of the collected perch samples could be considered to be affected by different doses of radionuclides due to radioactive contamination of freshwater ecosystems in the studied area after the ChNPP accident in 1986. Based on the available data [14], the Lithuanian and Latvian territories experienced a load of radioactive material that was at least several times lower than that experienced by the heavily polluted Belarusian and Ukrainian zones. Unfortunately, not enough samples of the perch population inhabiting the ChNPP cooler could be obtained to comprehensively study the possible effects of radiation and thermal pollution caused by the RBMK-1000 reactors. Nevertheless, there was an opportunity to study the possible anthropogenic impact on a perch population inhabiting Lake Drūkšiai. For more than 25 years, this water body served as a cooler of the INPP with two RBMK-1500-type reactors [79,80]. The INPP was established in the Lithuanian part of the coastal zone of Lake Drūkšiai and is located close to the Latvian and Belarusian borders [27]. The overall anthropogenic effect on perch genetics was studied because it was not possible to distinguish the impacts of the INPP, such as thermal pollution and radiocontamination, from the main pollution source of Lake Drūkšiai, including treated wastewater that was used for household needs in settlements like the town of Visaginas. It is necessary to consider that the Lake Drūkšiai ecosystem experienced many drastic changes before and after it was established as the INPP cooler [19,80]. Not all ecological changes were associated with exploitation of the INPP, as this water body experienced great morphological changes prior to 1976 [27]. During the period of operation of the INPP, one or both reactors were cooled, producing thermal pollution and emitting radionuclides into the environment [80]. The effect of radionuclides on pelagic organisms was quite low [79], but thermal pollution was responsible for fish reproduction failures [19] as well as the formation of warm and cold zones in this lake [27]. There were ecological differences between the fish, including perch, that inhabited the warm and cold zones in this lake. After the INPP was shut down at the end of 2009 [75], the zone of warmer water disappeared and the ecosystem experienced new ecological changes. This lake was oligo-mesotrophic before exploitation of the INPP, but now Lake Drūkšiai, the largest lake in Lithuania, has a mesotrophic status and some of its zones have become eutrophic [19]. Finally, attempts were also made to collect and study perch samples that could reveal the impact of various possible anthropogenic activities not associated with the INPP.

To sum up, the most representative samples (n = 19–30) were obtained from Lakes Cirīšu and Engure in Latvia, the Mozyr area in Belarus, and Lake Drūkšiai and the Neris River in Lithuania (Table 1), including sample (9b) collected in a segment of the river located close to the Belarus–Lithuania border. The Neris River flows through Vilnius and is contaminated by wastewater from industrial sources, a phenomenon that was reinforced after exploitation of the Astravets Nuclear Power Plant (ANPP) constructed less than 50 km from Vilnius on the bank of the Neris River. It is worth mentioning that there are plans to expand the ANPP by establishing an additional third reactor. Samples collected from the Elektrėnai Reservoir represented a water body that is used to cool the Lithuanian Thermal Power Plant (LTPP). Similar to Lake Drūkšiai, the Elektrėnai Reservoir was created by flooding other smaller water bodies. Finally, the Dotnuvėlė River is distinguished by marked chemical pollution, and the sample collected from the Akademijos Reservoir also represented a perch population profoundly affected by anthropogenic impacts. 

### 2.2. DNA Extraction, Amplification, and Sequencing

The genomic DNA of perch was extracted from frozen or ethanol-preserved muscle or fin clips following the universal and rapid salt-extraction method [81] with slight modifications. The mtDNA *ATP6* fragments were amplified with the help of newly designed primers: ATP-PCR-F (5′-CCCTAACGAGCCTACATCCC-3′) and ATP-PCR-R (5′-TGTAAGAGGTCAAGGGCTGG-3′). Briefly, available perch mtDNA sequences deposited in GenBank were retrieved and aligned, and conservative fragments with unchanging nucleotide positions suitable for primer design were chosen using Primer3 program [82]. 

Most polymerase chain reactions (PCRs) were performed in a total volume of 25 μL, consisting of 5 μL of perch genomic DNA, 12.5 μL of DreamTaq DNA polymerase mix (Thermo Fisher Scientific Baltics, Vilnius, Lithuania), 5.5 μL of nuclease-free water, and 1 μL of each primer. The other PCRs were performed in a total volume of 25 µL, including 2.5 μL of 10 ×Taq buffer with (NH_4_)_2_SO_4_, 2.5 μL of 25 mM MgCl_2_, 2.5 μL of dNTP Mix (2 mM), 0.2 μL of 5 U/μL 500 U Taq DNA polymerase (Thermo Fisher Scientific Baltics), 1 μL of each primer, 10.3 μL of water, and 5 μL of template DNA. Amplification started with an initial denaturation for 2 min at 95 °C; followed by 35 cycles of 30 s at 94 °C, 45 s at 52 °C, and 45 s at 72 °C; and a final elongation for 5 min at 72 °C. The PCR products were purified with the help of ExoI and FastAP enzymes (Thermo Fisher Scientific Baltics) and sequenced directly using the Big-Dye^®^ Terminator v3.1 Cycle Sequencing Kit and a 3500 Genetic Analyzer (Applied Biosystems, Foster City, CA, USA). 

### 2.3. Analysis of Intraspecific Genetic Data

Truncated *ATP6* region sequences—a 627 base pair (bp) fragment representing nucleotides 1–589 of *ATP6* and nucleotides 589–627 of cytochrome *c* oxidase subunit III (*cox3*)—were analysed to define different haplotypes using FaBox v. 1.5 [83]. Notably, the identified singletons were sequenced twice. Sequences representing newly obtained *ATP6* region haplotypes were deposited in GenBank under accession numbers OQ676936–OQ676948. The ClustalW algorithm [84] implemented in MEGA7 software [85] was used to align the sequences. The vertebrate mitochondrial code (transl_table = 2) derived from GenBank was used to translate the mtDNA *ATP6* region nucleotide sequences into amino acids. The Hasegawa–Kishino–Yano (HKY) evolutionary nucleotide substitution model [86] appeared to be most appropriate for phylogenetic analysis conducted using the maximum likelihood (ML) algorithm. Bootstrapping analysis with 1000 replicates was carried out to test the robustness of suggested phylogenies. The sequence (accession number MH301079) of yellow perch (*Perca flavescens* Mitchill) was selected as the outgroup sequence from GenBank. The haplotype network was created using the median joining (MJ) method implemented in NETWORK 10.2.0.0 software [87]. Related haplotypes in the haplotype network were attributed to the corresponding haplogroups. DnaSP v6 was used to estimate the number of polymorphic sites in DNA sequences (S); haplotype diversity (h) [88]; nucleotide diversity (π) [89]; average number of nucleotide differences (K) [90]; and selection coefficient (ω), calculated as the dS/dN ratio (non-synonymous substitution rate/synonymous substitution rate) to evaluate the possible effects caused by natural selection [91]. To evaluate the genetic differentiation of the examined perch samples, pairwise Φ_ST_ values were calculated using Arlequin v. 3.5.2.2 [92]. The statistical significance of the obtained Φ_ST_ values was tested by 10,000 permutations at the 95% confidence level. Principal coordinates analysis (PCoA) using Nei’s genetic distance [93] was performed using GenAlEx v. 6.502 [94].

### 2.4. Comparisons of Genetic Diversity Parameters between mtDNA ATP6 and D-Loop Regions

Trends of genetic variability based on the *ATP6* and D-loop regions were evaluated by comparing sequences of perch individuals collected from Lakes Cirīšu and Engure in Latvia and Lake Drūkšiai in Lithuania. This endeavour used the *ATP6* region data collected in this study and the mtDNA D-loop data obtained from a previous study [35]. Specifically, mtDNA D-loop region sequences of available perch from the Chernobyl area and Elektrėnai Reservoir were obtained during the present study using the same primers and PCR procedures as previously described [35]. For extended comparisons, additional information about the genetic variability of the same mtDNA regions was included using data available from GenBank for specimens inhabiting other water basins in Europe and Asia: AP005995, MZ461595, CM020933, MT410943, KM410088, LC495488, and AP018422. 

This study also used data reported by Ragauskas et al. [35] for the S, h, and π parameters, as well as SAMOVA data, extraction, and representation of the genetic diversity patterns in the studied macrogeographic area using the above-mentioned mtDNA regions. The values of these parameters were calculated by excluding all gaps in the mtDNA D-loop region. The distribution of the genetic diversity parameter (S, h, and π) values in the studied macrogeographic area are presented without standard deviations. Samples containing less than five individuals were not considered. Regarding the representation of S and π values on the map, five categories were distinguished based on minimum and maximum values of both mtDNA regions. Similarly, h values were represented with five categories, but the selected minimum and maximum values were 0 and 1, respectively. The established genetic groups ascribed using PCoA or SAMOVA were represented on the map by connecting dots among the same groups.

## 3. Results

### 3.1. Identification of Haplotypes and Haplogroups and Their Distribution Patterns

We identified a total of 13 different haplotypes among 193 samples from the macrogeographic area under study by comparing the sequences of the *ATP6* region (Figure 1 and Figure 2). It should be noted that as many as eight haplotypes (A1, A2, A3, B1, C1, C2, D1, and D2) were singletons and we identified haplotype D3 in two individuals (Appendix A). We discovered three singletons (A2, B1, and C1) in the Lake Cirīšu perch sample. Similarly, we found two singletons (A1 and A3) only in the Siesartis River perch sample. We detected singletons in Lithuanian (n = 108), Latvian (n = 40), and Ukrainian (n = 8) samples but not in Belarussian samples (n = 37) (Figure 1). Haplotypes A, B, C, and D had much higher frequencies compared to the previously mentioned haplotypes. The prevailing haplotype of Eurasian perch in the studied macrogeographic area was haplotype A (84 sequences, corresponding to 43.52%). The second most common haplotype was haplotype C, represented by 44 (22.79%) sequences. Finally, haplotypes D and B were represented by 28 (14.51%) and 27 (13.99%) sequences, respectively. Additional information about the frequencies of the most common haplotypes is presented in Appendix A. 

The nearest haplotypes were separated by 1–2 mutations in the haplotype network (Figure 1) and were attributed to four haplogroups designated with capital letters A, B, C, and D. Most of the point mutations in the amplified mtDNA fragment were *ATP6* transitions. We detected transversions in *ATP6* at position 319 (A→C in haplogroup B and A→T in haplogroup D) and in *cox3* at position 607 (G→C in haplotype D3). The ML dendrogram also supported the separation of four haplogroups (Figure 2). Based on the bootstrapping values, there were at least two distinct evolutionary lines (AC and BD), with one matriline represented by haplogroups A and C, and the other line represented by haplogroups B and D. In general, there were no clear patterns of haplotype distribution among the lakes, rivers, estuaries, and artificial reservoirs (Figure 2 and Appendix A), as we detected common haplotypes everywhere, and only singletons indicated some trends. Nevertheless, we detected the doubleton haplotype D3 only in the samples of two artificial reservoirs (the Akademijos and Elektrėnai Reservoirs). Additional parameters, such as the size, depth, and eutrophic status of the studied water bodies (Table 1), were not clearly linked to the distribution of haplotypes. However, we detected the common haplotype B and singleton B1 only in perch samples collected at altitudes that ranged from 80 to 160 m (Figure 2). 

Haplogroups A, B, C, and D were represented by 87 (45.08%), 28 (14.51%), 46 (23.83%), and 32 (16.58%) sequences, respectively (Figure 1). We detected representatives of all of these haplogroups only among the Neris River, Lake Cirīšu, and Ukrainian perch samples (Appendix A). We found the highest and lowest genetic variability in haplogroups A and D (represented by four different haplotypes each) and haplogroup B (represented by haplotypes B and B1), respectively. The results of the study (Figure 1 and Appendix A) showed that with some exceptions, the representatives of haplogroups A and C could be detected throughout the entire investigated macrogeographic area. The representatives of haplogroup B were more common in the southeastern part (Ukraine) of the studied macrogeographic area. Their presence decreased gradually in the northwestern part (the Baltic Sea), while haplogroup D showed the opposite tendency. Interestingly, only representatives of haplogroup B were not detected at lower altitudes (<80 m) (Table 1 and Figure 2). The representatives of haplogroups B and C were completely absent only in the Curonian Lagoon and Elektrėnai Reservoir, haplogroup D was absent in all Belarussian perch samples, and only haplotype D1 was detected in Ukrainian samples. Interestingly, there were no representatives of haplogroups B and D in Lake Žeimenys. This was the only case in Lithuania, as all other samples were representatives of either haplogroup B or D.

### 3.2. PCoA Results

Based on PCoA, the first and second principal components explained 68.8% and 25.2% of the genetic variation, respectively, and three genetic perch groups could be defined by this analysis (Figure 3). The first group (I) consisted of samples from the Curonian Lagoon, Elektrėnai Reservoir, Dotnuvėlė River (Akademijos Reservoir), and Lake Engure (Lithuanian and Latvian samples). The second group (II) also comprised Lithuanian and Latvian perch samples: it encompassed Lithuanian Lakes Drūkšiai and Žeimenys, the Neris and Siesartis Rivers, and Latvian Lake Cirīšu. Finally, group III was represented by Belarussian and Ukrainian samples. We concluded that the grouping of samples followed a geographic distance pattern, but with some specifics. For example, the first genetic component showed two distinct groups of Lithuanian and Latvian samples that belonged to the hydro-systems located in the continental eastern part (encompassing the Neris and Daugava Rivers) and the central-western part of the Nemunas River basin, together with coastal waters of the Baltic Sea, including the Curonian Lagoon and the Gulf of Riga, respectively. 

### 3.3. Quantitative Parameters of Genetic Variability and Genetic Differentiation among Perch Samples

The calculated parameters of genetic variability—h, K, S, and π—in different perch samples are presented in Table 2. All values of h, K, S, and π were 0 in the perch sample from the Meleshkovichi River channel. If we were to ignore this homogeneous sample, then the values of h would range from 0.382 ± 0.113 (Lake Žeimenys) to 0.786 ± 0.113 (Ukraine), the values of K would range from 0.38235 (Lake Žeimenys) to 2.03571 (Ukraine), the values of S would range from 1 (Lake Žeimenys) to 6 (Lake Cirīšu), and the values of π would range from 0.00061 ± 0.00018 (Lake Žeimenys) to 0.00325 ± 0.00055 (Ukraine). Consequently, the lowest and highest genetic variability were in Lake Žeimenys and Ukraine, respectively. 

Among the pairwise comparisons of the different perch samples, Φ_ST_ values ranged from −0.077 to 0.900 (Table 3). This indicated that there was no genetic differentiation between some compared samples (Φ_ST_ < 0.05); however, there were also samples that showed marked genetic differences from one another (Φ_ST_ > 0.25). Aside from the Meleshkovichi River channel sample, we found the most significant Φ_ST_ value (0.641; *p* = 0.001) by comparing the Curonian Lagoon and Lake Žeimenys samples. This result supported the PCoA result, as the distance between these two samples was the largest based on the first principal component (Figure 3). 

### 3.4. Protein Sequence Variability of the mtDNA ATP6 Region

The analysed 627 bp *ATP6* region consisted of one nucleotide before the first codon, 195 codons of the *ATP6* fragment, and two nucleotides (TA) that acted as a stop codon before the 13 codons that represented part of the *cox3* gene (Appendix A). The first nucleotide A after the mentioned TA sequence was used by both genes as a stop and start codon, respectively [69]. Of the 20 amino acids, only C was not coded by the studied fragment. We detected codons for K and D only once and in different genes. The most common amino acid was L (represented by 49/208 codons). Of note, L was translated by five out of six possible codons (we did not observe codon CTG). Similarly, the second most common amino acid, A (represented by 20 codons), was decoded by three out of four possible codons. Among the 193 perch *ATP6* region sequences, we found 13 different haplotypes containing two non-synonymous substitutions: one each in *ATP6* (codon ATT→GTT at position 84, leading to the I→V amino acid change in haplotype D1) and *cox3* (codon GCA→CCA at position 202, leading to the A→P amino acid change in haplotype D3) partial sequences (Table 4). Nucleotide substitution relative to the selected reference sequence represented by the most common haplotype A (OQ676936) also occurred at the first position of codon 171 in several haplotypes attributed to the B and D haplogroups, but this substitution did not code for a different amino acid. In most cases, we detected nucleotide substitutions in the third position of codons. Based on data obtained from GenBank, two additional *ATP6* region haplotypes could be detected outside the studied macrogeographic area. These two unique *ATP6* region haplotypes were represented by two specimens from China (AP005995 and MZ461595) and the perch mtDNA genome from Hungary (KM410088), which could be distinguished from haplotype A by two and four point mutations, respectively. Even so, there were no additional non-synonymous substitutions.

### 3.5. Natural Selection

The calculated dS/dN ratios for groups I, II, and III using the entire *ATP6* region as well as those without the *cox3* gene fragment are presented in Table 5.

All ω values were below 1 and generally indicated the effect of relaxed purifying selection. We observed a lower total dS/dN ratio for the *ATP6* partial sequence that for the entire studied *ATP6* region. This indicated stronger effects of purifying selection in the studied part of the *ATP6* gene than in the *cox3* gene. The lowest ω values in the *ATP6* region and partial *ATP6* sequence occurred in genetic group II, which included the majority of studied individuals (n = 101) and dominated the eastern Lithuanian and Latvian territories. Conversely, we detected carriers of haplotypes D1 and D3 that had non-synonymous substitutions between geographically distant territories (Ukraine and Lithuania). This could imply that natural selection might have had different effects on perch populations inhabiting northern parts compared to those in warmer parts of Europe.

### 3.6. Haplotypes and Haplogroups of mtDNA ATP6 and D-Loop Regions of the Same Specimens

The *ATP6* and D-loop sequencing data of the same individuals are presented in Appendix A. The results indicated that two above-mentioned mtDNA regions evolved at least partially independently and natural selection could have been responsible for these differences (Appendix A).

### 3.7. Comparisons of Trends of Genetic Diversity of Perch within the Studied Macrogeographic Area Using Different mtDNA Markers

Both mtDNA regions indicated that there were fewer haplotypes and haplogroups detected in the studied Belarussian perch populations than in the Latvian and Lithuanian perch populations (Figure 1, Appendix A). 

In general, the S values calculated using the *ATP6* marker were lower (min, most, and max: 0, 3–4, and 5–6 polymorphic sites, respectively) than those calculated using the D-loop region (min, most, and max: 1–2, 5–6, and 7–8 polymorphic sites, respectively) among the Lithuanian, Latvian, Belarussian, and Ukrainian samples (Figure 4). We found higher S values among the western part of Latvia (the Gulf of Riga and Lakes Engure and Babites) than in areas with a similar altitude (0–10 m) and the western part of Lithuania (the Baltic Sea, Curonian Lagoon, and Nemunas River). We calculated higher S values for perch samples from Lakes Engure and Drūkšiai, which are of similar size but differ in their altitude and depth (Table 1). We observed a trend of lower S values in Belarus using both markers. 

The h values calculated using both mtDNA markers in the studied macrogeographic area are presented in Figure 5. We found lower but quite similar general diversity in the *ATP6* region compared with the D-loop; most samples had values that ranged from 0.6 to 0.8 for both markers. We found higher h values among the western part of Latvia than in the elevated (altitude > 130 m) eastern part (Lakes Cirīšu and Sventes) of this country and the western part of Lithuania. The lowest diversity based on both markers was in the Meleshkovichi River channel sample. In Lithuania, the Elektrėnai Reservoir and Lake Plateliai could be indicated as hot and cold spots, respectively, as D-loop variability was highest and lowest, respectively, in these two samples. Interestingly, Lake Plateliai is more elevated (>140 m) than the surrounding western region near the Baltic Sea (0–10 m). Based on the *ATP6* region, the Desna River in Ukraine was a hot spot. When combining all Ukrainian samples, the h value (0.786 ± 0.113) was still higher than that of the combined Belarussian samples (0.611 ± 0.046) (Table 2). Conversely, Lake Žeimenys was a cold spot surrounded by an area with lower altitude (60–100 m) in the south (Figure 5) with similar h values, and an area of similar altitude (>130 m) but more than ten times larger h values than Lake Drūkšiai in the north.

Figure 6 shows the values of nucleotide diversity among the studied perch samples using different molecular markers. There were noticeably higher nucleotide diversity values using the D-loop marker (ranging from 0 to 0.006) than using the *ATP6* region (ranging from 0 to 0.004). The Desna River sample had the highest value for the *ATP6* region. When combining the Ukrainian samples, the obtained value was still the highest (0.00325 ± 0.00055) (Table 2). Based on the D-loop marker, the part of the Gulf of Riga containing Lakes Babites and Engure was a hot spot in the studied macrogeographic area. We observed the highest values in Lithuania among the Elektrėnai Reservoir, Neris River, and Lakes Drūkšiai, Dysnai, and Luodis. There were notable differences between the western and eastern parts of Latvia based on both markers. The western part of Lithuania also had lower genetic diversity than the western part of Latvia represented by territories around the Gulf of Riga. The Meleshkovichi River channel sample from Belarus had the lowest nucleotide diversity.

The established *ATP6* genetic groups (I, II, and III) (Figure 3) showed a non-accidental geographic distribution, as they encompassed previously ascribed groups I, II, and IV, respectively, established based on the D-loop marker (Figure 7). The obtained results fit the geographic distance as well as height above sea level patterns within and among the studied countries. The types of water bodies and specific features of particular locations, such as depth or trophic status (Table 1), had no obvious connection with these three perch genetic groups. In general, our results showed that *ATP6* could be quite a useful additional marker to study the phylogeographic tendencies of perch (Figure 4, Figure 5, Figure 6 and Figure 7). 

## 4. Discussion

### 4.1. Application of the ATP6 Region to Study Perch Population Genetics and Phylogeography

Different mtDNA regions as well as other types of molecular markers and their combinations, including the entire mtDNA genome, have different potentials to reflect distinct historic events as well as microevolution (see Appendix A). Therefore, it was reasonable to assess the potential of *ATP6* to study perch population genetics, including phylogeography and possible changes in genetic variation of perch populations related to anthropogenic activities in Europe. Our work provides new specific knowledge of the genetic variability of perch based on a mtDNA molecular marker that includes the majority of the *ATP6* gene and part of the *cox3* gene. We emphasised evaluating the possible effects of natural selection and trends of species genetic variation within a macrogeographic area encompassing the Black and Baltic Sea Regions. 

Thus far, the possible effect of natural selection specifically on the genetic variability of mtDNA *ATP6* in natural fish populations has been determined both interspecifically [76] and intraspecifically [77]. However, compared with marine fish species that have already been studied in a few scientific reports, there is a clear research gap that needs to be filled in this respect regarding freshwater fish species in Europe. The results (Table 5) of the current study suggest that the perch mtDNA *ATP6* region has been affected by relaxed purifying selection, and the purifying effect on the *ATP6* sequence was more pronounced compared to that on the entire studied mtDNA fragment. Consequently, in this regard, our results are similar to those reported by Deng et al. [77] regarding the intraspecific genetic variability pattern in a marine fish (*Ammodytes personatus* Girard) for which ω ranged from 0.0167 (lineage B) to 0.0384 (lineage A). Pichaud et al. [95] demonstrated that perch mtDNA gene expression, especially the NADH dehydrogenase subunit 4 (*nd4*) gene and variations representing adaptability of this gene, was highly affected by warmer temperatures. Our results (Table 5) indicate that the *ATP6* region could also be affected by natural selection within a larger geographic scale that profoundly differs in its climatic conditions.

The mtDNA genomes of perch populations representing haplogroup M of the D-loop profoundly differed in at least several mtDNA regions, including the *ATP6* gene, compared with their counterparts representing other haplogroups of the D-loop (Appendix A). Vasemägi et al. [46] reviewed the role of salinity, pH, and other possible important factors responsible for local adaptation and general genotype–phenotype links in perch, providing guidance for future studies. The results of the first investigation of Lithuanian and Latvian perch samples using the mtDNA D-loop region were published only in 2007 [96]. A short, summarised history for perch phylogeographic context is presented in Appendix A. The *ATP6* marker results revealed interesting but less complicated phylogeographic relationships among the studied perch populations (Figure 1, Figure 2, Figure 3 and Figure 7) compared with those revealed by the D-loop [35]. Figure 8 provides a summary of the major trends of genetic variability in the studied macrogeographic area. 

The three *ATP6* genetic groups (I, II, and III) (Figure 3) showed a non-accidental geographic distribution (Figure 7 and Figure 8). It is interesting that different haplogroups dominated within these three groups. Haplogroup D was most common within group I, haplogroup A dominated within the largest group II, and haplogroups B and C dominated group III, which was not homogeneous (Appendix A). The samples within groups I and II were not genetically differentiated (Table 3). Most pairwise comparisons among the samples revealed significant genetic differentiation (Φ_ST_ values ranged from 0.097 to 0.900) between representatives of the three distinct groups. Consequently, based on molecular data obtained using the *ATP6* marker, the established groups I and II indicated an obvious qualitative genetic difference between perch inhabiting the Nemunas and Neris River hydro-systems (Figure 7 and Figure 8), respectively. Group III encompassed samples from Belarus and Ukraine, thus discriminating them from the perch populations of the Baltic countries. Based on PCoA, Lithuanian and Latvian perch populations were genetically more closely related to the populations inhabiting the studied water bodies in Ukraine compared with those in Belarus (Figure 3). Therefore, we hypothesise that there have been postglacial connections between the current Dnieper, Daugava and Neris River basins, while the hydro-system of Belarus has remained more isolated, probably due to higher altitudes (Figure 2). The results revealed that microevolution of the mtDNA *ATP6* and D-loop regions was partly independent (Figure 8 and Appendix A). In future studies, it would be reasonable to use these mtDNA regions both separately and as composite *ATP6*–D-loop sequences. During their concatenated mtDNA investigation, Toomey et al. [38] found that π values in most European territories ranged from 0.0008 to 0.0009, but they identified hot spots in the Baltic Sea Region. The results of the current study agree with the finding of hot spots, but the π values were much higher (Table 2). Nevertheless, the values of this parameter we determined for the *ATP6* region were still lower than the values based on the D-loop (Figure 6). Other phylogeographic trends of both mtDNA markers are shown in Figure 4, Figure 5, Figure 6 and Figure 7 and Appendix A. The highest and lowest genetic diversity were in Latvia and Belarus, respectively (Figure 4, Figure 5 and Figure 6 and Appendix A). Perch populations in the western part of Latvia had richer genetic variability than their counterparts in the eastern part of this country and the western part of Lithuania.

We can conclude that the newly determined trends indicated by *ATP6* overlapped with most of the perch phylogeographic trends in Eurasia determined by sequencing other mtDNA regions [35,38,41,62], despite different sampling strategies and data analysis techniques (Figure 7 and Figure 8). The observed geographic distribution of the intraspecific genetic variation of perch mtDNA is poorly explained only by the currently existing natural barriers [38]. The water bodies, isolation by distance, and possibly altitude account for genetic differentiation among the studied perch populations in the eastern part of the Baltic Sea Region [35]. Finally, the results strongly support previous findings indicating an uneven distribution of genetic variability based on mtDNA-based analyses [38]. Indeed, the uneven distribution of species genetic variability could be detected on different geographic scales, and this could be explained by historical events within multiple refugia that in turn consisted of smaller refugia and different water basins [34,37,41,97]. The effects on the distribution of genetic diversity of postglacial freshwater fish species caused by both current and previously existing hydro-systems from the last glacial and Holocene periods, especially the flow of rivers in the Black and Baltic Sea Regions, requires more detailed research. Additional recommendations regarding future research of perch phylogeography are presented in Appendix A. 

### 4.2. Anthropogenic Context

Because of the recent challenges of the 21st century—such as the brutal war initiated by Russia in Ukrainian territory causing an economic and humanitarian crisis—the risk of radiocontamination is increasing, mainly due to NPPs located in the war zone (for example, the Zaporizhzhia NPP) [98]. In addition, the risk of possible anthropogenic contamination is related to the trend of building more NPPs in Europe, especially in Poland and Sweden, in the near future and various accidents associated with chemical pollution, including oil spills. From a scientific viewpoint, this creates interesting opportunities to comprehensively investigate changes and tendencies in genetic and epigenetic variability, as well as microevolution in action [99], caused by anthropogenic activities. The ChNPP and Fukushima disasters, as well as previous areas of nuclear weapons testing, serve as sites to evaluate the short- and long-term effects of radiocontamination on organisms, local ecosystems, and the environment. 

It seems that radiation has less of an effect on the genetic variability of aquatic organisms compared with terrestrial organisms [40,100], mainly due to the fact that plants are stationary and accumulate high doses of radiation, and mammals are the most radiosensitive class among animals [13]. In the Chernobyl area, the radioecological consequences were highest for fish species among all cold-blooded aquatic organisms [13]. mtDNA markers have been useful for assessing the mutagenic effect of ionising radiation, at least in one aquatic species [101]. Based on the current knowledge regarding the involvement of the *ATP6* gene in microevolution [102] and its interspecific [76] and intraspecific [77] variations in response to different environmental conditions, we expected to detect either greater intraspecific genetic differences in perch populations affected by pronounced anthropogenic activities, or at least differences of the same magnitude as what had been observed for the D-loop. Our results (Figure 1, Figure 4, Figure 5, Figure 6, Figure 7 and Figure 8, Table 2 and Table 3, and Appendix A) indicate that the summed anthropogenic effect on perch, including both radionuclides and thermal pollution affecting NPP coolers, could hardly be determined using the observable variation in the mtDNA *ATP6* region. Several generations have passed since the anthropogenic activities increased in the studied location, which could have concealed the increase in new point mutations. A similar trend was observed in previous research with perch using the mtDNA D-loop region [75]. Mutations might occur in the *ATP6* region because of specific anthropogenic activities, such as contamination of the environment by radionuclides, and could be detected in specimens of perch affected by anthropogenic activities (chemical and NPP pollution), but subsequent natural selection or reparation processes could eliminate new mutants from the population. The findings suggest that neither hypothesis H1 nor H2 could not be outright rejected, especially using more sensitive methods. However, their confirmation was not likely because the obtained results could be better explained by the perch phylogeographic trends observed in Lithuania, Latvia, and Belarus (Figure 1, Figure 2, Figure 4, Figure 5, Figure 6, Figure 7 and Figure 8). We can conclude that it is more likely that patterns of unique haplotypes or lower genetic diversity within and among particular territories of the studied macrogeographic area were mainly observed due to phylogeographic relationships among perch populations connecting them before or during the Holocene. The impact of NPP or TPP activities on the genetic makeup of contemporary perch populations could not be clearly demonstrated using spatial samples and the selected mtDNA marker.

To detect other possible, subtler transformations of genetic variability, such as changes in haplotype frequencies over different timescales of NPP exploitation, temporal samples should be studied [66,103]. In the case of Lake Drūkšiai, major temporal samples could be studied using available archival material or by collecting material for genetic studies from elder (≥20 years) perch individuals that still inhabit this lake. Perch age could be determined based on their otolith data [104,105] or even scales. Thus far, studies based on perch temporal sampling in areas encompassing NPPs have been conducted only using DNA microsatellites [49] and MHC genes [58]. Both studies investigated the Baltic Sea perch populations inhabiting the Forsmark Nuclear Power Plant (FNPP) area in Sweden. The first research based on DNA microsatellites revealed that genetic differentiation among temporal samples of perch as well as roaches (*Rutilus rutilus* L.) was better explained by isolation over time rather than anthropogenic activities caused by FNPP operation for more than two decades. On the other hand, MHC class IIβ genes experienced strong natural selection effects within the artificial cooler of the FNPP. Similar studies devoted to evaluating the anthropogenic impact of different NPPs on the genomes of fish located in brackish and freshwater ecosystems are required. It is also reasonable to consider expanded research of transcriptomic profiles of the affected and intact perch populations, as it has been shown that messenger RNA (mRNA) changes could appear due to attempts to domesticate wild perch [106]. In addition, transcriptional changes in the ovaries of perch from the Chernobyl area have been reported [9].

Single molecular markers, including whole mtDNA molecules, have a one locus limitation but are still useful as a first research step. Notably, the trends of genetic diversity revealed in the current work strongly support previous findings based on mtDNA-based analyses, and the results reported by Toomey et al. [38] based on DNA microsatellites representing eight loci were generally in agreement with the findings of the studied concatenated mtDNA fragment. Ultimately, hypotheses H1 and H2 require additional testing using more sensitive molecular markers, including whole genomes and transcriptomes of perch and additional species. Other fish species with different biologies and ecologies from perch might have different patterns and trends of their mtDNA *ATP6* genetic diversity within the studied macrogeographic area. Thus far, *ATP6* intraspecific research of various freshwater fish species in Europe is extremely limited [107]. Comprehensive investigations into the morphometric [60,66], genetic, and epigenetic [108] variability of perch and other fish in areas near the ChNPP as well as other NPPs and TPPs would fill many important scientific gaps [15,16,99], especially regarding fish microevolution that may be influenced by NPP and TPP effects.

We can conclude that the accumulated data and trends revealed in the current study could also be used in other important Anthropocene research areas and for practical recommendations. Major areas could include tracking changes in perch genetic diversity at particular sites due to natural or artificial changes, such as unintended oil spills, aquaculture, fisheries [35,38], and conservation, as well as the application of perch genetic authentication of geographic localisation [55]. The results from this study (Figure 7) support the recommendations to use, manage, and conserve perch resources in Latvia, Lithuania, and Belarus from an earlier study [35]. Thus far, there are no representatives of haplogroup M of the D-loop that also have unique *ATP6* region haplotypes in the studied macrogeographic area (Table 4 and Appendix A). Given that the newly built ANPP is located near the Lithuanian border, the accumulation and monitoring of genetic data based on mtDNA and nuclear markers of perch and other fish species from different sites of the Neris River should be investigated continually by following previous recommendations [99,109,110]. An investigation into the fragmentation of perch genetic variability caused by hydroelectric dams constructed in Latvia using DNA microsatellites was recently begun [25]. The study is going to be continued and expanded using mtDNA markers. Lastly, we recommend taking a cautious approach and avoiding the establishment of closed-type NPPs in the future [75,111]. There are alternatives to NPPs and other energy sources are currently available and there is no doubt that wildlife within NPP coolers experience harmful effects at higher levels than their intraspecific genetic diversity level [17,18,112].

## 5. Conclusions

The first comprehensive attempt to reveal major patterns and trends of the intraspecific genetic diversity of *P. fluviatilis* mtDNA *ATP6* within a macrogeographic area comprising the Black and Baltic Sea Regions showed less complicated, non-accidental, but similar phylogeographic relationships among perch populations compared with those reconstructed using the D-loop marker. PCoA revealed the existence of three perch genetic groups (I, II, and III) represented by different compositions of A, B, C, and D haplogroups specifically distributed in the studied area. Of note, the Lithuanian and Latvian perch populations could be discriminated from those inhabiting the heterogenous hydro-systems of eastern Belarus and Ukraine. Microevolution of the mtDNA *ATP6* and D-loop regions has been at least partially independent. Although the current study provides the most recent data and trends od the genetic and protein variability of perch based on the *ATP6* region, our hypotheses H1 and H2 regarding the impact of anthropogenic activities on the genetic structure of perch remain to be comprehensively tested. As the next major step, we recommend performing genomic and transcriptomic investigations of perch populations affected by NPPs or TPPs, as research within the anthropogenic context requires applying the most sensitive methods.

## Figures and Tables

**Figure 1 animals-13-03057-f001:**
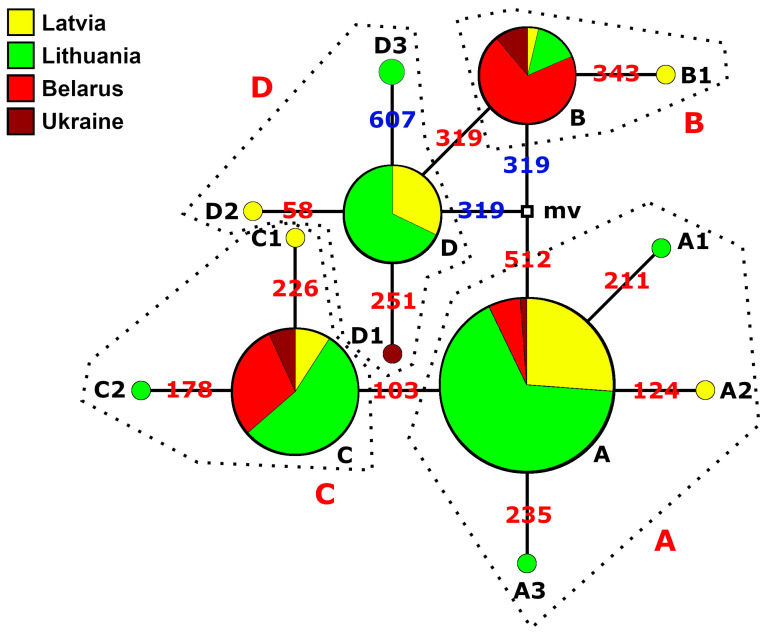
Haplotype–haplogroup network based on 13 perch haplotypes. The black and red letters indicate haplotypes and haplogroups, respectively. The circle radius (from 1 to 84) shows the number of specimens with distinct haplotypes. The numbers indicate positions of mutations. The numbers in red and blue represent transitions and transversions, respectively, while mv is a median vector (a computer-generated sequence that presumably existed in the past or still does). All distinct haplogroups are marked with dotted lines. Different countries are represented by samples as described in Table 1.

**Figure 2 animals-13-03057-f002:**
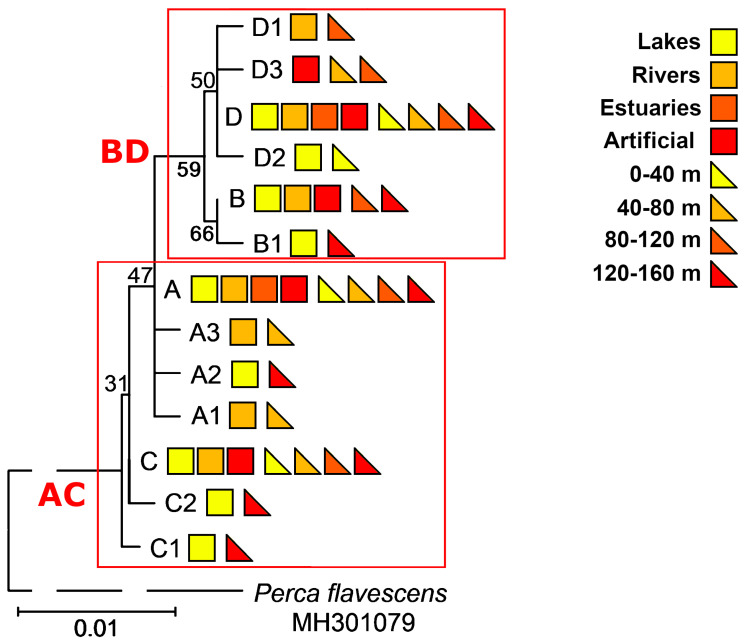
Maximum likelihood (ML) dendrogram of 13 perch haplotypes representing at least two distinct evolutionary lines: AC and BD. The haplotype attributed to the related species, *Perca flavescens*, represents the outgroup. The numbers next to the branches indicate the bootstrapping support values. The dashed line does not represent evolutionary distance. The coloured squares and triangles represent the type of water body and altitude, respectively.

**Figure 3 animals-13-03057-f003:**
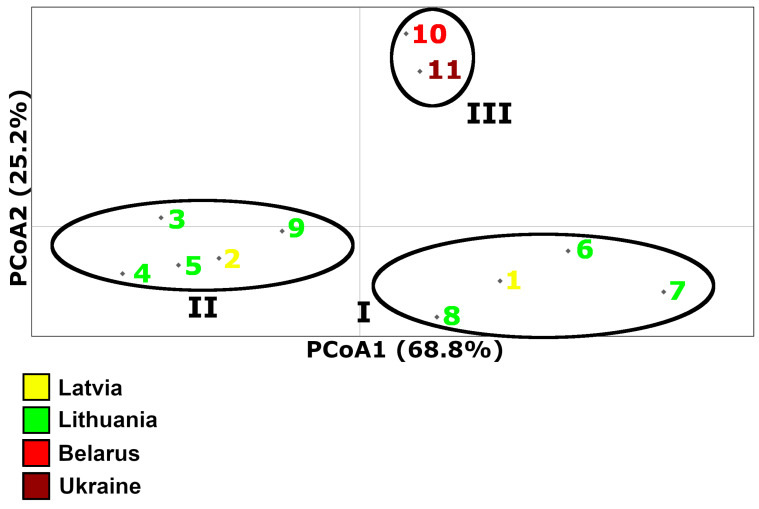
Principal coordinates analysis (PCoA) of Latvian, Lithuanian, Belarusian, and Ukrainian perch samples. There are three genetic groups: I, II, and III. Codes: 1—Lake Engure; 2—Lake Cirīšu; 3—Lake Drūkšiai; 4—Lake Žeimenys; 5—Siesartis River; 6—Dotnuvėlė River; 7—Curonian Lagoon; 8—Elektrėnai Reservoir; 9—Neris River; 10—Belarus; 11—Ukraine.

**Figure 4 animals-13-03057-f004:**
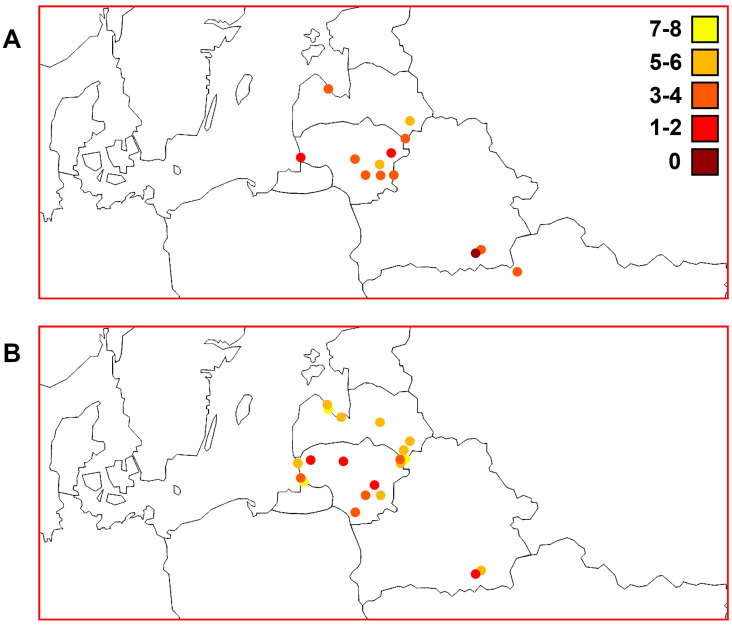
Distribution patterns of polymorphic sites (S) among perch samples from Lithuania, Latvia, and the transboundary region between Belarus and Ukraine. (**A**) Mitochondrial DNA (mtDNA) *ATP6* region genetic diversity (see Table 2). (**B**) mtDNA D-loop region genetic diversity based on data reported by Ragauskas et al. [35].

**Figure 5 animals-13-03057-f005:**
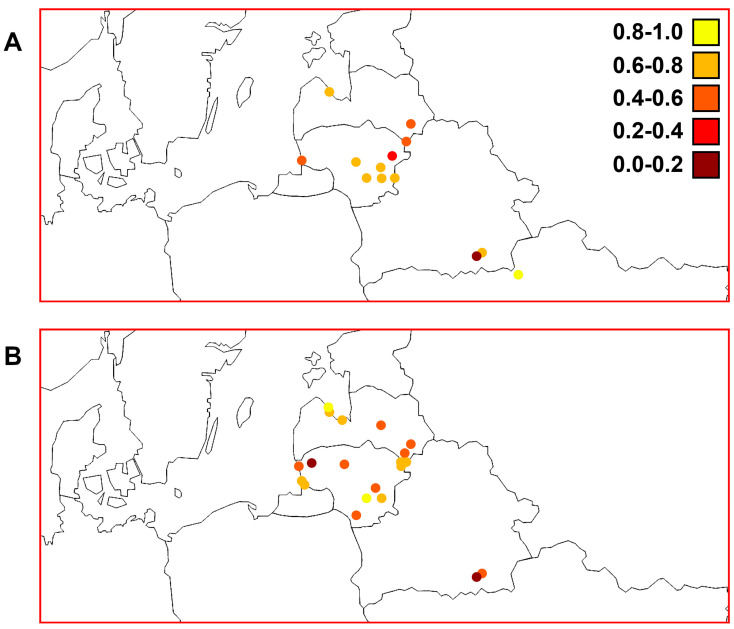
Distribution patterns of haplotypic diversity (h) among perch samples from Lithuania, Latvia, and the transboundary region between Belarus and Ukraine. (**A**) Mitochondrial DNA (mtDNA) *ATP6* region genetic diversity (see Table 2). (**B**) mtDNA D-loop region genetic diversity based on data reported by Ragauskas et al. [35].

**Figure 6 animals-13-03057-f006:**
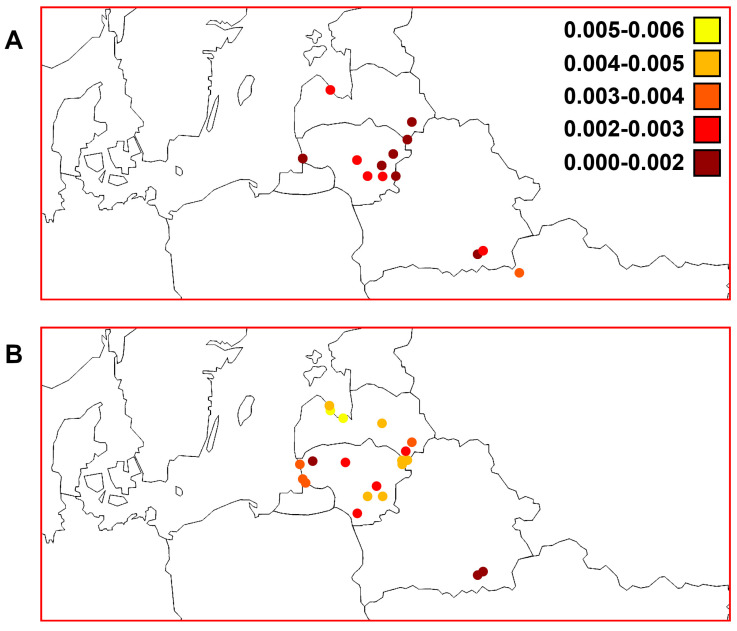
Distribution patterns of nucleotide diversity (π) among perch samples from Lithuania, Latvia, and the transboundary region between Belarus and Ukraine. (**A**) Mitochondrial DNA (mtDNA) *ATP6* region genetic diversity (see Table 2). (**B**) mtDNA D-loop region genetic diversity based on data reported by Ragauskas et al. [35].

**Figure 7 animals-13-03057-f007:**
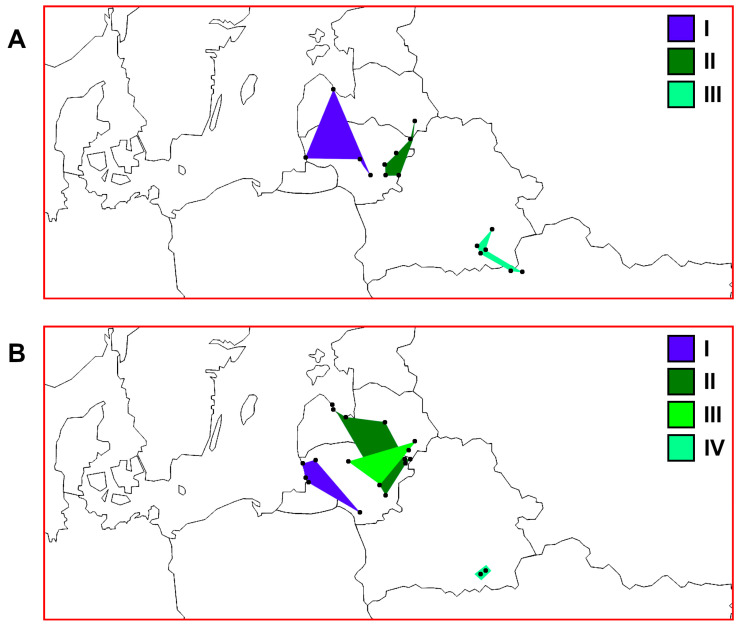
Distribution patterns of perch maternal genetic groups in Lithuania, Latvia, and the transboundary region between Belarus and Ukraine. (**A**) Mitochondrial DNA (mtDNA) *ATP6* region genetic groups based on PCoA (see Figure 3). (**B**) mtDNA D-loop region genetic groups based on SAMOVA [35].

**Figure 8 animals-13-03057-f008:**
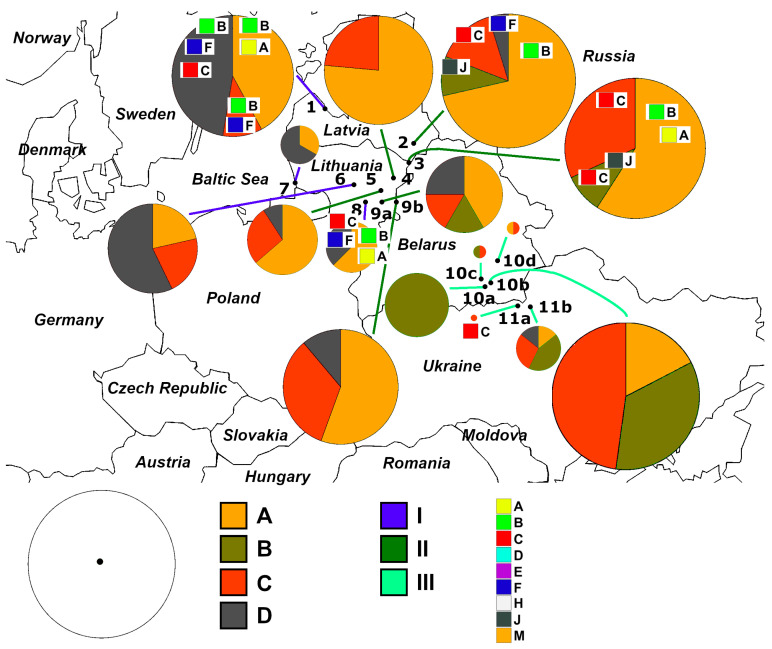
The distribution of haplogroups of the *ATP6* region among the Latvian, Lithuanian, Belarusian, and Ukrainian perch populations. The link between distinct haplogroups of the *ATP6* and D-loop regions in a few studied samples (see Appendix A) are shown as additional smaller squares with letters (representing all D-loop haplogroups) on circles. The circle radius indicates the sample size: the smallest sample size (n = 1) is indicated as a dot while the largest sample size represents 23 individuals. Distinct haplogroups are shown in different colours. Line colours represent perch genetic groups that were distinguished based on principal components analysis (see Figure 3). The samples are ordered as described in Table 1.

**Table 1 animals-13-03057-t001:** Location, basin, sampling time, and sample size of perch used for this study.

Code	Location	Basin	Sampling Time	Sample Size
Latvian samples				
1	Lake Engure North: 57.258, East: 23.117 Altitude 3.2 m, average depth 0.4 m, max depth 2.1 m, 4046 ha Eutrophic	Baltic Sea	2015	19
2	Lake Cirīšu North: 56.133, East: 26.966 Altitude ~140 m, average depth 4.1 m, max depth 10 m, 630.6 ha Eutrophic	Daugava River	2011	21
Lithuanian samples				
3	Lake Drūkšiai North: 55.621, East: 26.605 Altitude 141.6 m, average depth 7.6 m, max depth 33.3 m, 4487 ha Mesotrophic	Daugava River	2009	22
4	Lake Žeimenys North: 55.287, East: 26.06 Altitude 138.3 m, average depth 6.9 m, max depth 23.5 m, 443.8 ha Mesotrophic	Žeimena River	2018	17
5	Siesartis River North: 55.293, East: 24.909 Altitude 60 m	Siesartis River–Šventoji River	2015	11
6	Dotnuvėlė River (Akademijos Reservoir) North: 55.4, East: 23.85 Altitude 66.2 m, average depth 1.8 m, max depth 5.9 m, 35.4 ha Eutrophic	Dotnuvėlė River	2014	14
7	Curonian Lagoon North: 55.35, East: 21.197 Altitude 4 m, average depth 3.8 m, max depth 5.8 m, 158,400 ha Eutrophic	Baltic Sea	2017	6
8	Elektrėnai Reservoir North: 54.758, East: 24.669 Altitude 94.9 m, average depth 7.2 m, max depth 31 m, 1264 ha Eutrophic	Strėva River	2017	8
9a	Neris River (Baltalaukis) North: 54.862, East: 25.401 Altitude 100 m	Neris River	2009, 2010	12
9b	Neris River (Buivydžiai) North: 54.836, East: 25.734 Altitude 105 m	Neris River	2018	18
Belarusian samples				
10a	Meleshkovichi River channel North: 51.91, East: 28.968 Altitude 100–120 m	Pripyat River	2015	10
10b	Mozyr North: 52.06, East: 29.257 Altitude 120–140 m	Pripyat River	2015	23
10c	Lake Aleksino North: 52.107, East: 28.744 Altitude 100–120 m	Pripyat River	2015	2
10d	Berezina River North: 52.65, East: 29.772 Altitude ~116 m	Berezina River–Dnieper River	2015	2
Ukrainian samples				
11a	Chernobyl cooling pond North: 51.385, East: 30.137 Altitude 110 m, average depth 6 m, max depth 18 m, 2290 ha Eutrophic	Pripyat River–Dnieper River	2012	1
11b	Desna River North: 51.077, East: 30.889 Altitude ~111 m	Desna River	2012	7

**Table 2 animals-13-03057-t002:** Quantitative parameters of intraspecific genetic variability of the studied Lithuanian, Latvian, Belarussian, and Ukrainian perch samples.

Code—Sample	n	h	K	S	π
1—Lake Engure	19	0.667 ± 0.066	1.35673	4	0.00216 ± 0.00025
2—Lake Cirīšu	21	0.562 ± 0.126	1.06667	6	0.00170 ± 0.00050
3—Lake Drūkšiai	22	0.593 ± 0.089	0.89177	4	0.00142 ± 0.00035
4—Lake Žeimenys	17	0.382 ± 0.113	0.38235	1	0.00061 ± 0.00018
5—Siesartis River	11	0.764 ± 0.107	1.16364	5	0.00186 ± 0.00049
6—Dotnuvėlė River	14	0.703 ± 0.095	1.56044	4	0.00249 ± 0.00037
7—Curonian Lagoon	6	0.533 ± 0.172	1.06667	2	0.00170 ± 0.00055
8—Elektrėnai Reservoir	8	0.607 ± 0.164	1.32143	3	0.00211 ± 0.00057
9—Neris River	30	0.669 ± 0.060	1.16782	3	0.00186 ± 0.00028
10—Belarus	37	0.611 ± 0.046	1.49550	3	0.00239 ± 0.00012
10a—Meleshkovichi River channel	10	0	0	0	0
10b—Mozyr	23	0.648 ± 0.055	1.47036	3	0.00235 ± 0.00022
11—Ukraine	8	0.786 ± 0.113	2.03571	4	0.00325 ± 0.00055
Overall ^a^	189	0.720 ± 0.021	1.37690	12	0.00220 ± 0.00011
Overall ^b^	193	0.721 ± 0.021	1.37817	12	0.00220 ± 0.00011

^a^ Without two small Belarusian samples. ^b^ All samples.

**Table 3 animals-13-03057-t003:** Genetic differentiation of perch samples. Pairwise Φ_ST_ comparisons and the significance of these values (*p* < 0.05) are presented below and above the diagonal, respectively. Significant Φ_ST_ values are shown in light grey, while significant values after Bonferroni correction are presented in dark grey.

Code	1	6	7	8	2	3	4	5	9	10a	10b	11
1 (I)	-	0.653	0.601	0.706	0.015	0.002	0.001	0.020	0.050	0.001	0.014	0.263
6 (I)	−0.041	-	0.633	0.457	0.010	0.001	0.001	0.029	0.034	0.001	0.026	0.354
7 (I)	−0.047	−0.077	-	0.591	0.014	0.001	0.001	0.009	0.021	0.001	0.019	0.272
8 (I)	−0.064	−0.033	−0.005	-	0.116	0.014	0.009	0.081	0.054	0.001	0.067	0.387
2 (II)	0.156	0.208	0.323	0.080	-	0.311	0.390	0.614	0.475	0.001	0.066	0.097
3 (II)	0.262	0.298	0.467	0.228	0.005	-	0.647	0.733	0.246	0.001	0.095	0.047
4 (II)	0.331	0.389	0.641	0.344	−0.001	−0.026	-	0.880	0.110	0.001	0.021	0.010
5 (II)	0.190	0.224	0.377	0.133	−0.025	−0.037	−0.049	-	0.431	0.001	0.100	0.110
9 (II)	0.097	0.132	0.257	0.190	−0.011	0.015	0.055	−0.012	-	0.001	0.132	0.242
10a (III)	0.505	0.480	0.682	0.646	0.653	0.743	0.900	0.752	0.607	-	0.001	0.007
10b (III)	0.164	0.159	0.271	0.143	0.081	0.066	0.156	0.077	0.035	0.498	-	0.454
11 (III)	0.023	0.003	0.052	−0.001	0.101	0.157	0.269	0.112	0.031	0.383	−0.032	-

Code: 1—Lake Engure; 2—Lake Cirīšu; 3—Lake Drūkšiai; 4—Lake Žeimenys; 5—Siesartis River; 6—Dotnuvėlė River; 7—Curonian Lagoon; 8—Elektrėnai Reservoir; 9—Neris River; 10a—Meleshkovichi River channel; 10b—Mozyr; 11—Ukraine.

**Table 4 animals-13-03057-t004:** Protein sequence variability of different haplotypes of the *ATP6* region. Nucleotide substitutions and amino acid changes are marked in red letters. Codons representing *cox3* gene or having non-synonymous substitutions are shown shaded in grey. The last codon showed no differences in all haplotypes.

Haplotype	Codons																
	*ATP6*															*cox3*	
	19	34	41	59	68	70	75	78	84	106	114	121	125	155	171	202	208
A	ATC I	GGC G	CTA L	TAT Y	TCC S	AAT N	GTT V	TGA W	ATT I	CTA L	GAA E	CGC R	TTG L	CCA P	TTA L	GCA A	CCC P
A1						AAC N											
A2			CTG L														
A3								TGG W									
B										CTC L					CTA L		
B1										CTC L	GAG E				CTA L		
C		GGT G															
C1		GGT G					GTC V										
C2		GGT G		TAC Y													
D										CTT L					CTA L		
D1									GTT V	CTT L					CTA L		
D2	ATT I									CTT L					CTA L		
D3										CTT L					CTA L	CCA P	
New1 AP005995										CTT L			TTA L				
New2 KM410088			TTA L		TCT S							CGA R		CCC P			

**Table 5 animals-13-03057-t005:** Calculated mitochondrial DNA (mtDNA) *ATP6* dS/dN ratios of perch genetic groups distinguished based on principal components analysis (see Figure 3).

Genetic Group	n	dS/dN	
		*ATP6* Region (627 bp)	*ATP6* Partial (589 bp)
I	47	0.088768	0
II	101	0	0
III	45	0.089767	0.090444
Total	193	0.065331	0.03314

## Data Availability

The mtDNA *ATP6* region sequences retrieved in the current investigation were submitted to the NCBI GenBank database under accession numbers OQ676936–OQ676948.

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
