# Peer review of "Trends of Eurasian Perch (Perca fluviatilis) mtDNA ATP6 Region Genetic Diversity within the Hydro-Systems of the Eastern Part of the Baltic Sea in the Anthropocene"

_animals, 2023, doi:10.3390/ani13193057_

Round 1
Reviewer 1 Report
The genetic diversity and population structure of Perca fluviatilis were comprehensively analyzed by mtDNA ATP6 gene sequences in the manuscript, and the conclusions in this study were also compared with those of based on D-loop sequences in previous researches. The fundings are helpful for identifying the impacts of human activities on genetic variation of Perca fluviatilis in the eastern part of the Baltic Sea region. Although the research paper is seriously written, it appears so verbose. Please simplify the manuscript as much as possible. Moreover, why choose ATP6 gene as the DNA marker? It isn’t well explained in the “Introduction” section. Just because other common molecular markers have already been used? The data analyzed in this study didn’t seem to be the ATP6 gene sequences. Actually, authors used the partial sequences of ATP6 and COIII genes, and it also confuses me. Why not amplify the complete sequence of ATP6 gene? Therefore, I think “ATP6 genetic diversity” in the title is also inappropriate.
It can be accepted after minor revision.
Author Response
Answers in document.

Reviewer 2 Report
This manuscript is an extensive study of a gene in in the mitochondrial genome of perch populations.
I have two major concerns. First the structure of the manuscript makes it very challenging to read. It is fairly too long given that it is a study of a single mtdna gene. The hypotheses are not presented until the methods, referred to again only late in the discussion, and tl Ben then as h and h2. I think the organization and focus of the study can be improved considerably.
second, there is too much over interpretation of these results. The a analysis of dn/ds ratios is inconclusive relative to the populations. For a study of a single mtdna gene sequence ther is significant ant liberty taken about the impact on population distribution unions without much discussion of the limitations.
why did the authors not use a structure analysis, common for these types of data and more powerful for examining admixture of genotypes?
I think this is a good and worthwhile study that could be refocused, written more concisely, and interpreted more fairly.
Author Response
Answers in document.

Round 2
Reviewer 2 Report
Thank you for the revisions. I find them acceptable.